# Single-Cell DNA Methylation Analysis in Cancer

**DOI:** 10.3390/cancers14246171

**Published:** 2022-12-14

**Authors:** Hannah O’Neill, Heather Lee, Ishaan Gupta, Euan J. Rodger, Aniruddha Chatterjee

**Affiliations:** 1Department of Pathology, Dunedin School of Medicine, University of Otago, Dunedin 9016, New Zealand; 2School of Biomedical Sciences and Pharmacy, College of Health, Medicine and Wellbeing, The University of Newcastle, Callaghan, NSW 2308, Australia; 3Hunter Medical Research Institute, New Lambton Heights, NSW 2305, Australia; 4Department of Biochemical Engineering and Biotechnology, Indian Institute of Technology Delhi, New Delhi 110016, India; 5School of Health Sciences and Technology, University of Petroleum and Energy Studies (UPES), Dehradun 248007, India

**Keywords:** DNA methylation, single cell, cancer

## Abstract

**Simple Summary:**

Cancer is a distinctly difficult disease to treat on account of the diverse cell populations/subpopulations that comprise a tumour. Such cells harbour varying genetic and epigenetic states, which contributes to their oncogenic phenotype. Of note to this review is the epigenetic modification DNA methylation. Aberrant DNA methylation is a well-explored contributor to oncogenic phenotype. Traditionally, thousands of cells within a tumour have been sequenced together, giving rise to averaged methylation profiles. With the emergence of single-cell sequencing technologies, the methylome of individual cells can now be acquired. This technology will have important research and clinical implications, such as informing our current understanding of cancer biology, discovery of novel biomarkers, and less invasive tests.

**Abstract:**

Morphological, transcriptomic, and genomic defects are well-explored parameters of cancer biology. In more recent years, the impact of epigenetic influences, such as DNA methylation, is becoming more appreciated. Aberrant DNA methylation has been implicated in many types of cancers, influencing cell type, state, transcriptional regulation, and genomic stability to name a few. Traditionally, large populations of cells from the tissue of interest are coalesced for analysis, producing averaged methylome data. Considering the inherent heterogeneity of cancer, analysing populations of cells as a whole denies the ability to discover novel aberrant methylation patterns, identify subpopulations, and trace cell lineages. Due to recent advancements in technology, it is now possible to obtain methylome data from single cells. This has both research and clinical implications, ranging from the identification of biomarkers to improved diagnostic tools. As with all emerging technologies, distinct experimental, bioinformatic, and practical challenges present themselves. This review begins with exploring the potential impact of single-cell sequencing on understanding cancer biology and how it could eventually benefit a clinical setting. Following this, the techniques and experimental approaches which made this technology possible are explored. Finally, the present challenges currently associated with single-cell DNA methylation sequencing are described.

## 1. Introduction

Epigenetics has been described as “the study of changes in gene function that are mitotically and/or meiotically heritable and that do not entail a change in DNA sequence” [1]. In essence, the epigenome consists of numerous types of modifications to DNA, which act to regulate the genome [2]. The impact of epigenetics on the cancer genome has been referred to as “the most obvious source of dark matter” [3], exhibiting how elusive the field once was in its contribution to the cancer genome. Epigenetic mechanisms have now been recognised as contributors to the acquisition of cancer hallmark capabilities [4]. Perhaps the most well-explored epigenetic modification, DNA methylation (DNAme), can lead to oncogenic phenotypes when abnormal changes take place [5]. Through transcriptional regulation, altering genomic stability and induction of mutational events to name a few, DNAme heavily influences oncogenic phenotype [6]. Studying such marks and their contribution to oncogenesis is difficult, however, as tumours tend to be highly heterogeneous entities. Cellular methylomes are influenced by cell types and states, of which there are many in a single tumour. Fortunately, recent years have seen the introduction of single-cell sequencing (SCS), a technology that permits the analysis of single-cell methylomes so that comprehensive analyses of epigenetic heterogeneity are now possible.

This review will first discuss the importance of DNAme in relation to cancer, then discuss how single-cell DNAme analysis might advance clinical outcomes and further our understanding of cancer biology. The review will then concentrate on the technology and experimental methods that made this sequencing possible. The discussion of bioinformatic methods for analysing this data is followed by a breakdown of the present difficulties in the realm of this currently developing technology.

### 1.1. DNA Methylation

Epigenetic modifications are described as chemical modifications to the DNA that cause a heritable phenotype without changing the primary DNA sequence [7]. One of the main epigenetic alterations is DNA methylation (DNAme), a stable yet reversible mark entailing the addition of a methyl group, typically at the fifth position of a cytosine residue (5-methylcytosine, 5mC) adjacent to a guanine residue (CpG site) [8]. A total of 28.7 million CpG sites are present in the human genome; these CpGs are often found concentrated together, forming CpG islands (CGIs), which are regions of the genome with a higher frequency of CpGs than expected [9]. Usually, these CGIs are present in promoter regions of genes, and the presence or lack of DNA methylation influences the transcriptional activity of the associated gene. In most cases, the presence of DNA methylation (DNAme) at CGIs represses transcription, whereas DNAme absence is observed in transcriptionally active genes [10]. These regulatory properties of DNAme are crucial in numerous fundamental biological processes throughout the lifespan, including cell-cycle control, cell fate decisions, X-chromosome inactivation, genomic imprinting, embryonic development, chromosomal stability, and transposable element silencing [11,12,13,14]. Considering the above, it is not surprising that aberrant DNAme patterns are implicated in many diseases, including Alzheimer’s [15], cardiovascular diseases [16], and, of interest to this review, all types of cancer [17]. Furthermore, aberrations in enzymes associated with DNAme, such as ten-eleven translocation proteins, have been identified as cancer hallmarks [18], as well as intermediate states of 5mC, such as 5-hydroxymethylcytosine (5hmC) [19,20]. 

### 1.2. Single-Cell DNA Methylation

Traditionally, due to technical limitations, large numbers of cells have been combined and analysed as one to obtain insight into DNAme patterns. For conciseness, this practice will be referred to as “bulk sequencing” for the remainder of the review. When combining thousands of cells for analysis, an averaged methylome profile of all the cells will be produced. While this technique has been excellent in advancing our knowledge of the role DNAme plays in disease, there are some areas of research that bulk sequencing is not capable of reaching. For example, small subpopulations with distinct methylomes within tumours are present with greater metastatic capabilities; however, their methylomes will be masked by the many other cells present in the tumour [21].

Recent years have seen the emergence of single-cell technologies, allowing for the analysis of single cells’ epigenomes (Figure 1). In essence, this means cells with differing methylomes reflecting different cell types/states in a tumour can now be identified and appreciated as differing components of the tumour, rather than grouped together as one. This allows for analyses, such as characterisation of rare subpopulations and cell lineage tracing, which provide insight into the epigenetic regulation of heterogeneous populations of cells.

## 2. Single-Cell Methylome Profiling in Cancer

Collectively, the genome, epigenome, and transcriptome of cancer cells are grossly dysregulated. The exploitation of pathways required in normal biological processes results in the initiation, progression, dissemination, and metastasis of cancer. As a result of inactive tumour suppressor genes or overactive oncogenes, processes such as apoptosis, cell proliferation, and DNA repair mechanisms are interrupted [22], while others, such as angiogenesis, replicative immortality, and the ability to evade growth suppression, are influenced [23]. Epigenetic modifications have been implicated in cancer predisposition, early tumorigenesis events, metastasis, and therapeutic resistance [24]. 

A recurrent challenge faced while studying most types of cancer is its heterogenic nature. Heterogeneity refers to the state of being diverse. In the context of cancer, this can reflect the diversity between individuals (inter-individual heterogeneity), between primary tumours and their metastases (inter-tumoural), and between cells within the same tumour (intra-tumoural). Tumours are comprised of numerous types of cells, including the malignant and healthy cells from the tissue of origin, fibroblasts, immune cells, nerves, and subpopulations of tumour cells [25]. Single-cell sequencing provides a tool for understanding the complex dynamics and heterogeneity that encompasses each tumour. 

### 2.1. Single-Cell DNA Methylation in Cancer Initiation and Progression

The diversification of cells begins with cancer stem cells from the tissue of origin. These cells have self-renewal capabilities and multi-lineage differentiation, which encourage tumorigenesis and heterogeneity [26]. Cancer stem cells are known to drive initiation and progression, as well as encourage disease relapse [27]. Famously, cancer stem cells have been difficult to isolate due to their similarities to normal tissue-of-origin stem cells and their scarcity. The emergence of SCS means it is now possible to isolate cancer stem cells and study them to a higher resolution, without noise from other cells’ epigenomes influencing the data. One study was able to identify a small number of tumour-initiating cells in human glioma through the use of SCS [28], albeit with single-cell RNA sequencing; however, the same principles could be achieved with single-cell methylome techniques. This group was able to evidence that glioblastoma development occurs along with conserved neurodevelopmental gene processes, as well as potential sources of resistance to some immunotherapies in these cancer stem cells. 

Tumour cells heavily influence their surrounding environment, inducing chemical, molecular and physical changes. The hallmark features of the tumour microenvironment include blood vessels, immune cells, stromal cells, and extracellular matrix [29]. Following Darwinian processes, tumour cells are continuously interacting and being influenced by the tumour microenvironment, driving branched tumour evolution of cells with selectively advantageous traits to survive [30]. Branched tumour evolution describes when distinct subclones within the tumour evolve in parallel, leading to a considerable amount of diversity between subclones [31]. Phenotypically, this evolution often contributes to evading the immune system, angiogenesis, and avoiding cellular growth-suppressing processes, encouraging tumour progression [32]. In consequence, a highly heterogeneous tumour is formed. Therefore, when combining a large population of tumour cells for analysis, the true individual cellular methylomes are averaged by neighbouring cells. As such, it is expected that cells within a tumour will exhibit great variation, encouraging the need for higher resolution techniques, such as single cell. One group utilised single-cell DNA methylome data to identify the lineage tree of subpopulations in human chronic lymphocytic leukaemia [33]. The lineage tree was created based on the epigenetic aberrations of the cells, showing different chronic lymphocytic leukaemia lineages that were discriminatively influenced and expelled from lymph nodes by ibrutinib, one of the treatments for chronic lymphocytic leukaemia.

The differentiation of cancer stem cells, followed by interactions with the tumour microenvironment driving branched tumour evolution are key events that produce the heterogeneity observed within and between tumours. The use of SCS technologies will not only augment our current understanding of these events, but possibly introduce new pathways for investigation as to how tumours initiate and progress. 

### 2.2. Single-Cell DNA Methylation in Metastasis 

Cancer dissemination and subsequent metastasis to new niches around the body is one of the leading causes of death in cancer patients [34]. Circulating tumour cells (CTCs) are highly tumorigenic cells that can lead to metastasis and, due to the low number of CTCs in patient blood, the epigenetic landscape of CTCs remains less characterised [35]. Previously, the low DNA content meant bulk sequencing was not an option for the analysis of CTCs. Using SCS to analyse CTCs provides opportunity to investigate the epigenetic processes that may drive metastatic dissemination of cancer cells and subsequent invasions at new locations around the body. SCS was used to show that CTCs from breast cancer patients showed a focal hypomethylation at a number of stem cell gene promoters, increasing the expression of pluripotency networks [36].

As well as studying CTCs, being able to isolate and analyse subclonal populations which have evolved from the primary to metastatic tumour sites may give insight into the pathways which are exploited for this process. One study used SCS on two secondary tumours (metastatic breast and metastatic castration-resistant prostate cancer), examining methylation patterns on epithelial–mesenchymal transition (EMT)-associated genes to predict metastatic potential [37]. This identified differentially methylated patterns between the two entities, showing that the miR200 feedback loops, involved in inducing epithelial differentiation in the EMT process, were differentially regulated. Another group reconstructed the lineage within a colorectal cancer patient between the primary tumour and its associated metastases and found global DNAme was stable during metastasis, with no DNAme alterations in EMT-related genes prior to or post metastasis [38]. This group also showed a sub-lineage within the primary tumour was detected in the lymph node and liver metastases of the patient, suggesting these metastases had a common origin.

### 2.3. Single-Cell DNA Methylation in Cancer Therapy

The role DNAme plays in relation to cancer initiation and progression has made DNAme and its regulators a well-explored target for cancer therapy [39]. Furthermore, studies have been able to predict responses to therapy through the use of DNAme biomarkers. For instance, glioblastoma patients with methylated *MGMT* promoters responded better to alkylating drugs than individuals without methylation [19]. Different subpopulations may have these methylation alterations at these important loci as a result of the branched tumour evolution outlined earlier, and SCS has the ability to uncover such populations. Furthermore, aberrant DNAme accumulation in cancer stem cells was shown to affect self-renewal capabilities, differentiation, multi-drug resistance, and metastasis processes in cancer stem cells [40]. As previously mentioned, cancer stem cells make up a minor population of the tumour, with far greater capabilities of responding to radio- and chemotherapies [26]. SCS could, therefore, enable the detection of cancer stem cells, as well as the discovery of novel biomarkers for therapeutically resistant cells. SCS has been used in a number of papers to look into issues that are clinically significant. For instance, it was utilised to compare the tumour microenvironment before and after immunotherapy in basal cell carcinoma [41]. This provided insight into the regulatory networks controlling cellular response to therapy, in particular, some overlapping regulation of therapy-responsive T cells. Moreover, different epigenomic subpopulations have been shown to vary in their response to targeted therapy, with certain subpopulations showing greater resistance to imatinib [42]. Understanding the molecular heterogeneity of the individual’s tumour could also be used to inform oncologists on which currently available treatments could be combined for the most effective treatment.

In a clinical context, it has also been proposed that SCS could provide noninvasive tests by taking liquid biopsies containing CTCs at different time points throughout a patient’s therapy [43]. Multiple studies have found that CTCs had over 50% of the same mutations as the primary tumour in lung cancer [44] and colon cancer [45]. Therefore, they could be extracted from blood and analysed to track the evolution of the primary tumour and adjust treatment accordingly. Another study found there was a substantial amount of methylome heterogeneity amongst distinct CTCs from the same patients [37]. This could lead to personalised signatures in these CTCs for patient stratification and treatment selection. Circulating tumour DNA (ctDNA) is tumour-derived DNA that has become free in the blood [46]. While ctDNA has also been used to extract epigenetic signatures from liquid biopsies for similar purposes to CTCs, CTCs represent still intact tumour cells, which are likely to give more comprehensive information regarding the tumour it disseminated from [47].

Single-cell techniques are most commonly used for RNA analyses to profile gene expression rather than DNAme analyses. A typical cell contains approximately 10–30 pg of RNA, which is then reverse transcribed into cDNA and amplified [48]. Conversely, the starting material for single-cell DNAme analyses starting material is only the double-stranded DNA (~6 pg maximum) [49]. The native DNA then requires manipulation, such as bisulphite conversion or enzymatic treatment prior to amplification; therefore, there is a high risk of DNA loss [50]. As such, the disparity between the number of publications of the two techniques can likely be attributed to the significantly lower starting material of DNAme and the library preparation techniques required and technical challenges associated with single-cell DNAme experiments (Table 1). Nonetheless, the epigenetic influence on oncogenic phenotypes and cancers’ inherent heterogeneity is a well-appreciated phenomenon, which warrants further investigation in the single-cell context. Moreover, DNAme is a stable epigenetic mark that is likely to represent a sustained signature in contrast to gene expression changes, which are more variable and represent dynamic changes in the cell. With the above points considered, single-cell DNAme sequencing is an enticing direction for the field, despite the challenges currently being faced.

## 3. Single-Cell DNA Methylation Sequencing Techniques

The recent ability to study populations at the single-cell level has come from technology enabling the isolation of single cells from a population, as well as improved amplification steps and growing bioinformatics tools.

### 3.1. Isolation of Single Cells

The first step in single-cell analyses is the isolation of the single cells from their population. There are a number of methods that rely on either the physical properties (non-affinity methods) of cells or cellular characteristics (affinity methods) [56]. Currently, the most frequently used technique for single-cell DNAme analyses is fluorescence-activated cell sorting (FACS). As the name implies, cells are labelled with fluorescent probes that conjugate to surface markers on target cells, which are subsequently run through cytometry. Here, they are exposed to a laser and given a negative or positive charge based on fluorescence and separated into their respective tubes [56]. Typically, in single-cell DNAme analysis, the cancer cells would undergo live—dead staining, as it will be a heterogeneous population of cells, with no single cell type being targeted [57]. However, other probes for cell-specific markers are available. FACS is able to isolate cells based on both surface markers, as well as physical aspects, such as size and granularity. All this being said, FACS does present with limitations. FACS requires a starting cell suspension containing roughly 10,000 cells, meaning samples with low cell counts will have to be isolated using a different method or cultured until the appropriate number of cells are present. Further, initially viable cells can be easily damaged during the cytometry process, rendering them no longer viable. A similar method to FACS is magnetic-activated cell sorting (MACS), which follows similar principles to FACS but uses magnets attached to the antibodies for the sorting of cells. While MACS can be four to six times faster than FACS [58], it also tends to be harsher on cells due to the magnetic nature of the process. This can increase cell lysis during the isolation process, posing an issue if the initial starting number of cells was already low. Both MACS and FACS have a high throughput of >1000 cells per run, making them efficient techniques [56]. Furthermore, fluorescence-activated nuclei sorting was utilised in the first described combinatorial indexing strategy applied to single-cell DNAme analyses, sci-MET [59]. This method involves cells being sorted into a 96- or 384-well plate, each well containing a unique indexed adapter. Following this, cells are pooled, redistributed, and introduced to a second index such that the likelihood of a cell having the same unique combination of indexes is low. While this method admittedly resulted in lower per-cell coverage percentages, coverage was sufficient for cell-type discrimination and allowed for higher throughput than other methods. 

Another method for isolating single cells from a heterogeneous population is laser capture microdissection (LCM). This method involves directing a laser at the sample under an inverted microscope to manually isolate single cells. This method would be preferable to FACS/MACS in the instance of low starting cell counts, such as solid tumour samples/fixed in formalin and embedded in paraffin (FFPE) samples. LCM does not destroy neighbouring cells/tissues [60], preserving neighbouring cells and leftover tissue, whereas FACS/MACS requires dissociation of the tissue. In a study that profiled the DNA methylome of lung cancer CTCs, LCM allowed for highly accurate isolation of the CTCs from the likes of immune cells, without compromising cell integrity [54]. There are two major limitations to LCM; this includes needing a highly skilled technician who is capable of identifying cells through visual inspections, as well as possible damage to the DNA as a result of the UV laser. Similar to LCM is a technique called micromanipulation, which also involves using an inverted microscope to manually isolate single cells. Typically, this technique uses a tool such as a micropipette to isolate the cells. This technique is preferred to LCM when isolating live cultures, rather than fixed tissue. However, the throughput of micromanipulation is far lower than that of LCM. Albeit both LCM and micromanipulation have lower throughputs than the aforementioned FACS and MACS methods (<100 cells per run) [61]. 

Finally, the most recent advancement in single-cell isolation is the use of microfluidics. Microfluidic chips contain hundreds of channels ranging from 10 to 100 μm to fit single cells. A number of microfluidic principles can then be used, such as hydrodynamic cell traps and oil droplets in water isolation [61]. Of the currently available techniques, the latter is the most frequently used, as the oil encapsulation of the cell reduces the chance of contamination, while also having the highest throughput (1000–10,000 cells per run) of the currently available techniques. Microfluidic cell isolation techniques have not largely been used in single-cell DNA methylation (scDNAme) analyses, with only a few to date [62,63], and none have yet been described in the cancer context. Although starting volume of cells required is small, with fast processing and high sensitivity making it a great candidate for isolating cells in cancer contexts [61], a reliable protocol for DNAme analysis has not been well established. In future, this method may become favourable for the analysis of single-cell methylomes once the technology is available and reliable; however, for now, it is not at a stage where this is possible.

Essentially, FACS/MACS both have higher throughput than LCM and micromanipulation; however, the latter techniques may be more fitting if one was investigating the likes of fixed tumour tissue. Considerations to be made for the choice of single-cell isolation methods and experimental techniques have been depicted in Figure 2.

Of note, a future direction for single-cell epigenomics which may render the cell isolation step unnecessary are spatial analyses. This has been performed in spatial transcriptomics, where cell types are assigned by gene expression in their exact locations on the histological section [64]. This in situ sequencing preserves the spatial context of cells within the tumour microenvironment [65]. While SCS is capable of identifying different cell types and states present in the tumour, it requires the dissociation of tissues and, therefore, spatial context is lost. No methods for spatial methylome analyses have yet been described; however, one spatial epigenomic technique was recently published, Epigenomic MERFISH [66]. This technique successfully profiled histone modifications marking active and silent promoters, as well as putative enhancers in single cells, while preserving their spatial context. 

### 3.2. Experimental Approaches

Once single cells have been isolated, several DNAme analyses can be employed dependent on the research goals (Table 2). Initial experiments employed the use of bisulphite conversion, the gold-standard technique in bulk analyses, allowing for single-base resolution. This technique involves treating the DNA with sodium bisulphite, which, in turn, converts unmethylated cytosines into uracils and, subsequent to PCR, an unmethylated cytosine would present as a thymine. Alternately, methylated cytosines will not be converted by sodium bisulphite treatment; therefore, they still present as cytosines post-PCR [67]. This technique involves denaturing the DNA in a high concentration of sodium bisulphite salt, low pH, and high temperature [68]. These conditions often lead to high levels of DNA degradation; hence, it is often referred to as a harsh process. The first technique used for single-cell DNAme sequencing used a bisulphite conversion-based approach, this being single-cell reduced representation bisulphite sequencing (scRRBS) in 2013 [69]. RRBS involves the use of both bisulphite conversion and a restriction enzyme (MspI) for digestion and DNA size selection to obtain DNAme information from subsets of the genome where most DNAme occurs. This method was adapted for single-cell sequencing by combining all five steps of RRBS (cell lysis, MspI digestion, end-repair/dA-tailing adapter ligation, and bisulphite conversion) into a single tube reaction [70]. This meant there was very little chance for DNA loss other than from bisulphite conversion. Improvements to the scRRBS technique have resulted in a number of RRBS-based single-cell methods. Quantitative RRBS (Q-RRBS) utilises unique molecular identifiers (UMIs) to eliminate duplication-induced artefacts, which tends to be more severe with smaller inputs, such as single cell [71]. Microfluidic diffusion-based RRBS (MID-RRBS), to name another, involves the use of a microfluidic device for diffusion-based reagent exchange, allowing for conversion with the least amount of DNA loss [62]. RRBS-based methods offer a lower sequencing cost than whole genome approaches, since it concentrates on only particular portions of the genome. Cost is particularly important in the single-cell context, as each cell’s genome requires sequencing, making it relatively expensive. The scRRBS method is limited in that it only measures 10–15% of CpG sites in the genome, as well as having bias sequence selection, as the restriction enzymes only cut at specific sites. Reflecting this, CpGs without the enzyme restriction site and non-CpG are missed when using this technique. Additionally, this method takes approximately 3 weeks to complete and requires high molecular biology skills [70].

The second technique, introduced in 2014, was also bisulphite conversion-based, single-cell genome-wide bisulphite sequencing (scBS-seq) [73]. This technique covers a greater proportion of CpGs in the genome compared to scRRBS. However, a different set of CpGs is covered in each cell, making comparison of epigenetic states difficult. Although post-bisulphite adaptor tagging (PBAT), which performs the bisulphite conversion before library amplification rather than after, has been implemented in these methods to retain as much DNA as possible, ineffective template recovery in subsequent library preparation is still common. While traditional bisulphite sequencing provides coverage of >90% of the 28.7 million CpGs in the genome, the scBS-seq method using PBAT yields only several million, ranging from 5 to 50% [57]. This renders the data sparse and relatively uneven across the genome. As a result of the need to sequence the entire genome, scBS-seq is also more expensive than other approaches. As an extension of these bisulphite-based methods, several adapted methods have emerged, the previously mentioned sci-MET being one, in which bisulphite-based methods were utilised in unison with combinatorial indexing to allow for higher throughput [59]. Single-nucleotide methylcytosine sequencing (snmC-seq) is another, in which the reaction is carried out within the nucleus rather than within the cell lysate, and involves the multiplexing of reactions leading to large-scale cell-type classification [77]. Moreover, a single-cell transposable element sequencing technique has been recently introduced, scTEM-seq, which utilises transposable elements (SINE-Alu) as surrogates for predicting global methylation of single cells [55]. This method drastically reduces the sequencing demand for single-cell DNA methylation analysis and allows thousands of scTEM-seq libraries to be pooled for sequencing.

Single-cell techniques have also emerged that do not require bisulphite conversion of the DNA but, rather, rely on restriction enzymes for the identification of methylated CpGs. Single-cell methylation-sensitive restriction enzyme sequencing (scMSRE) [76] and single-cell AbaSI sequencing (scAba-seq) [74] are examples of these. These methods cost less than conversion-based methods and are more efficient. scAba-seq also has the ability to distinguish 5hmC from 5mC residues, a favourable trait, as 5hmC has been shown in several studies to be implicated in oncogenic phenotypes [42,43,44]. However, these techniques have far lower coverage than scRRBS and scBS-seq, as they are confined to only cleaving at methylation present at restriction sites and would, therefore, be more appropriate in targeted/site-specific studies. Another non-bisulphite conversion-based technique often used in bulk DNAme analyses is methylated DNA immunoprecipitation sequencing (MeDIP-seq) [78]. Monoclonal antibodies are raised against the 5mC modification, followed by enrichment of methylated fragments during immunoprecipitation, which can subsequently be sequenced. The lowest minimum required input thus far for MeDIP-seq is 0.5 ng (~50–100 cells), through the use of a microfluidic-based MeDIP protocol [79]. A “carrier strategy” has also been implemented, in which a carrier, such as chemically modified peptides, increases the immunoprecipitation efficacy, allowing for ~50 cells to be sequenced [80]. To date, these are the lowest input MeDIP-seq protocols have achieved; as such, this technique has not yet been utilised in a single-cell context. Although, as methods emerge to continue improving immunoprecipitation efficacy, a single-cell protocol may develop. 

The above methods are common for single-cell DNAme analyses; however, it is becoming increasingly popular to use multi-omic single-cell methods. These methods are capable of integrating multiple different omics, including proteomes, genomes, epigenomes, and transcriptomes, from the same cell to give comprehensive insights into how the omics are all inter-related [81]. This allows for parallel profiling of multiple layers in single cells to identify causal relationships between epigenome regulation and gene expression. This is a favourable technique, as lack of methylation does not always infer gene expression and vice versa. These techniques involve the separation of genomic DNA (gDNA) from mRNA, often using Smart-seq2 to obtain transcriptome information, followed by single-cell bisulphite methods (scM&T-seq [82]), scRRBS [83], and smartRRBS [84]) for methylome information. Although this technique is beneficial for inferring a correlation between gene promoters and expression, DNAme contributes to oncogenic phenotypes in more ways than just activation or repression of genes [85]. This includes regulating noncoding RNAs, as well as providing biomarkers for clinical purposes. For example, the hypermethylation of the *Vimentin* promoter in colorectal cancer cells compared to adjacent healthy colon cells did not result in a change in expression; however, the presence of this methylation was heavily associated with the stage of the tumour and the likelihood of it metastasising [86]. Multi-omic methods extend further than just the methylome and transcriptome. Single-cell nucleosome occupancy and methylation sequencing (scNOME-seq) [87], for example, allows for the analysis of both the methylome and chromatin accessibility of a single cell. In fact, single-cell nucleosome, methylation, and transcription sequencing (scNMT-seq) [88] has been developed for the parallel profiling of chromatin accessibility, DNA methylation, and transcriptome of a single cell. Such multi-omic methods allow for the exploration of inter-relations between the epigenetic layers of a cell. 

Multi-omic approaches also certainly come with their own limitations. The same limitations from the mono-omics apply (explored earlier); however, the genomic DNA now needs to be separated from the RNA, as well as perform the other omics methods, such as scRNA sequencing. This can make the process far more time consuming than it already is and requires an even greater level of technical expertise. 

### 3.3. Third-Generation Sequencing Techniques for Single-Cell DNA Methylation

The era of third-generation sequencing is now fast approaching. Whereas second-generation sequencing consists of amplification and subsequent sequencing of DNA, third-generation sequences native DNA strands [89]. Two well-known techniques include Nanopore and PacBio HiFi sequencing, with methylation inference-based ionic current variations or fluorescence events, respectively [90,91].

PacBio observes, in real time, a DNA polymerase synthesising a DNA strand, incorporating fluorescently labelled nucleotides [92]. The colour of the pulses indicates the identity of the base, while the kinetics (how long it takes the polymerase to incorporate the base and how long it takes to go between adjacent bases) is affected by modifications of the DNA, such as methylation and the context of the base [90]. Flusberg et al. showed the time it takes the polymerase to add the corresponding base is increased when the native base is methylated [93]. Methylation of cytosines has a slightly more subtle kinetic signature compared to the likes of adenine methylation (observed in plants) and, as such, a method termed Aggregate on Intervals method was suggested by Suzuki et al., which combines neighbouring CpGs kinetic information to infer methylation status of areas such as repetitive regions, promoters, and CpG islands [94]. However, more recently, PacBio has introduced circular consensus HiFi sequencing, in which the same DNA strand is sequenced multiple times in series [95]. This means, while the kinetic change for one methylated cytosine may be subtle, multiple passes over the methylated cytosine will result in an aggregation of the signals giving a clearer kinetic signature [92]. Essentially, reading the sequence multiple times over improves sequence accuracy and observing kinetic signatures multiple times allows for more accurate methylation calls. Additionally, Tse et al. created a neural network model in which, for every consecutive CG in a read, a feature vector is produced with the kinetics (pulse widths and inter-pulse durations) in a 16kbp window around that CpG site [96]. This vector is fed into a neural network and produces an output with the probability of methylation. This study was able to show that the overall methylation levels deduced from HiFi sequencing were greatly correlated with those by bisulphjite sequencing (r = 0.99; *p* < 0.0001) [96].

Oxford nanopore technology (ONT) is the other commonly used third-generation sequencing platform. Nanopore entails the passing of native DNA through a pore and measuring the ionic fluctuations. Different ionic fluctuations correspond to different bases, and any alterations present on the DNA, such as DNAme, will also alter the signature [97]. A number of bioinformatic tools have been developed to infer methylation, the most frequented being DeepSignal [98], signalAlign [99], Nanopolish [100], and DeepMod [101]. ONT have already been utilised in the cancer context, with one group identifying marked demethylation and tumour-specific insertions of the transposable element LINE-1 between paired tumour and non-tumour liver tissue, exhibiting the ability of ONT to provide insight to transposable element mobilisation and the cancer epigenome [102]. The initially high error rate of ONT rose questions around the accuracy of the libraries, as many loci would be missing the necessary signals for methylation analysis. However, in the most recent release, the sequencing accuracy is >99.99% (Q50) at 20× coverage, a comparable score to next-generation sequencing (NGS), which is the current standard [103]. 

The ability to sequence native single DNA strands without the need for chemical conversion or restriction enzymes (indirect methods of measurement) would reduce numerous challenges currently being faced in scDNAme analyses. PCR amplification bias, particularly in GC-rich contexts, which are often the areas of interest, would no longer be a point of issue. 

In theory, PacBio and ONT technologies appear to be enticing new technologies to implement. However, specific PacBio/Nanopore hardware is needed to perform such sequencing, which, as with all new technologies in their infancy, is relatively expensive [104]. This means the accessibility of these technologies is restricted. In addition to this, single cells have ~6 pg of DNA as input, which is a fraction of the 400 ng minimum DNA input for PacBio and ONT [105,106]. The DNA cannot be amplified prior to long-read sequencing, as this would remove the methylation marks. As such, single-cell DNAme analyses are not yet possible. Despite these limitations, it is anticipated that the technologies will improve in accuracy and input needs as they advance, making them more inexpensive and eventually applicable in the setting of single cells.

## 4. Single-Cell DNA Methylation Bioinformatic Analyses

The process of single-cell DNAme bioinformatic analysis involves preprocessing the raw data, performing imputation and normalisation, followed by downstream analyses to identify the likes of differentially methylated regions (DMRs), clusters, and epigenetic lineage trees [107]. While there are often overlaps in the bioinformatic analysis process between single-cell and bulk data, unique challenges arise when analysing single-cell data. The computational overheads of single-cell methylome data are far greater than that of bulk methylome sequencing, with far noisier and more variable data [108]. While traditional bulk DNAme sequencing has a relatively standardised bioinformatic pipeline, single-cell DNAme does not. This affects reproducibility if labs are using custom scripts to preprocess their data. Below will explore each step typically taken in the bioinformatic analysis of scDNAme data, as well as currently available tools which have been designed specifically for scDNAme data (Table 3).

### 4.1. Preprocessing

The purpose of preprocessing is to transform raw reads into non-bias data capable of producing biologically relevant findings [118]. The preprocessing of single-cell DNAme data has many parallels with bulk DNAme data; however, it presents distinct challenges inherent to the nature of single-cell sequencing. Preprocessing begins with demultiplexing the raw reads to distinguish the individual cells, as single-cell experiments usually pool barcoded single cells together for high-throughput sequencing [50].

#### 4.1.1. Trimming

Following this, indexed reads undergo adapter and quality trimming using software such as Trimmomatic [119] or Trim Galore! [120]. Depending on the technique that was used, different trimming will be performed. For scBS-seq, a PBAT protocol is implemented as it allows for a very small amount of starting DNA. However, a heavy 5′ bias has been identified, with biased reads exactly proportionate to the length of the oligo used [121]. As such, in scDNAme libraries, it has been suggested that, as well as typical adapter and quality trimming, the first N (oligo length) bases at the 5′ end of a read should also be trimmed [121]. In paired-end libraries, however, such trimming at the 5′ end leads to what is known as dovetailing, where the ends of the paired-end libraries will extend past the other. Previously, such reads were considered discordant. However, in light of the new scDNAme-seq challenges, Bismark implemented a new function: dovetail, allowing for dovetail reads to be considered viable [122]. 

For scRRBS, Trim Galore! has an RRBS function as used in normal RRBS analysis. This function removes 2 bp from the 3′ end of each read to remove the “filled in” cytosine position near the MspI digestion site [120]. As of the time this is being written, there are no extra precautions that have been identified as needing rectifying through bioinformatic analysis as a result of scRRBS.

Once trimming is complete, FastQC can be used to determine whether any adapter contamination or low-quality sequences are still present. If libraries are still of poor quality, they may need to be excluded in further analysis. The second source of possible contamination which may interfere with alignment rates is species contamination. Therefore, post-trimming, a FastQ Screen should be performed to ensure no cross-contamination. A FastQ Screen takes the libraries and aligns them against multiple genomes to ensure little to no alignment against other species’ genomes, which may otherwise indicate contamination [123]. Recommended species’ genomes for this approach may include mouse, rat, *E. coli*, PhiX, and Lambda, plus any other species that may have the potential for cross-contamination from the lab in which the libraries were prepared [123]. Once such screening processes have been completed, if low alignment rates are still observed, then adapter/species contamination can be confidently ruled out as possible contributors to the low alignment. 

#### 4.1.2. Genome Alignment

Once cells pass the quality check from FastQC and FastQ Screen, libraries are then aligned to a reference genome to indicate genomic locations of CpG sites, typically using Bismark [122]. In both single-cell and bulk data, the bisulphite sequence reads tend to be AT-rich as a result of the C to T conversions. Inevitably, reads appear more similar in sequence and can be mapped to similar genomic locations, despite having come from different areas. There are multiple aligners available that aim to resolve this problem. However, the genomic coverage of single-cell data is far more sparse genomic coverage than that of bulk data, making the alignment step even more difficult [107]. The optimal method of alignment for single-cell libraries is still up for discussion and varies largely between studies. Depending on the library preparation used (i.e., PBAT, RRBS, and MSRE), a number of issues can arise. In the case of single-cell bisulphite conversion-based methods, a PBAT paired-end preparation protocol is used. Despite this, the libraries should be treated as single-end nondirectional libraries during genome alignment [57]. This is because the multiple rounds of preamplification lead to hybrid fragments, which cannot be aligned as valid paired-end alignments, as well as original top, complementary to original top, original bottom, and complementary to original bottom strands, which, therefore, require nondirectional alignment [57]. 

It has also been suggested that reads which do not map with typical end-to-end alignment could be mapped using local alignment [124]. Despite local alignment being a less stringent method, in the instance of less reliable reference genomes or, this case, sparse data, the local alignment method can be used. Local alignment allows for soft clipping of reads, which entails the clipping of reads at either end of the sequence that do not align to the reference genome [124]. Therefore, only a fraction of the read must align, whereas the ends can be soft clipped. As one would expect, this now means once uniquely aligned reads may be able to align to multiple areas of the genome, such as repetitive elements, as displayed by Felix Krueger in an investigation into the effects of soft clipping [124]. Considering bisulphite conversion typically involves the use of a three-letter alphabet, which already reduces complexity and consequently leads to less unique alignments [125], the use of soft clipping in local alignment further reduces the likelihood of obtaining uniquely mapped reads. Krueger found the alignment rates increased when using local alignment instead of end-to-end; however, almost all the coverage differences could be attributed to mapping at repetitive regions, such as satellites, simple repeats, and low-complexity regions [124]. This suggested that reads were trimmed until they fitted one of these repetitive regions, despite the fact they most likely did not originate from that genomic region. As a result, local alignment has been recommended as a last resort to improve mapping efficiency. The choice between using a local alignment method or end-to-end essentially becomes a trade-off between having higher mapping efficiencies yet less confidence that these are the true genomic locations.

Chimeric reads also impact on the low genome alignment in single-cell conversion-based methods. It has been shown that a substantial amount of chimeric reads arise due to recombination of genomic proximal sequences with microhomology regions [126]. In light of this, Wu and colleagues suggested aligning the paired-end sequences in the usual PBAT mode (—pbat), then writing out the reads from each pair that do not align [126]. With the unmapped reads for read 1, alignment should be carried out in single-end mode under the “—pbat” function again, while read 2 should be run in single-end directional mode. The three final alignment files for these steps should then be methylation-extracted before being combined. This can salvage data, as chimeric/hybrid reads can result from the PBAT protocol and, therefore, mapping the unmapped reads from each pair as singles rather than pairs ensures Bismark does not disregard reads because they do not line up with their mate. While this method can be good if alignment is still very low, typically, libraries will be aligned as previously described in single-end nondirectional mode.

For samples which have low mapping rates and reads are removed as quality control, however, these parameters need to be adjusted to single-cell data, as they inherently have lower levels due to genomic coverage sparsity [107]. The cut-off for mapping rates and reads may vary depending on the median values of these parameters in the sample. For example, Liu et al. removed cells with <500,000 reads and <50% genomic mapping rates. The cytosine CpG and non-CpG contexts were also filtered, as is conducted in bulk, with the parameters mCC < 0.03, mCG > 0.5 and mCH < 0.2 [127]. The number of CpG sites covered also acts as a quality metric, deducing the libraries’ diversity. Traditional WGBS typically covers > 87% of CpG sites in the genome [128], whereas this number varies across single-cell DNAme techniques. For example, snmC-seq2 has an average coverage of 30.8 ± 7.5% [75], compared to sci-MET average of 1.1 ± 0.9% [59].

The preprocessing of scDNAme data is a crucial step that has bottleneck effects on the downstream analyses if performed incorrectly. Currently, there are very few standardised pipelines available for the preprocessing step of single-cell methylome data. This means much of the data going into downstream analysis pipelines have varying methods of preprocessing, largely influencing the metadata outcomes and, consequently, biological interpretations. One pipeline, SINBAD [113], has been created, which includes a preprocessing module. SINBAD is a flexible computational tool that can be used for the complete analysis of single-cell methylome data, from raw reads through to cell clusters and biomarker statistics. Conversely, Methyl Star [112] is a pipeline created solely for preprocessing single-cell methylome data. This pipeline incorporates frequently used next-generation sequencing tools, such as FastQC, Trimomatic, Bismark, and Methimpute, into one user-friendly interface, with variations for both expert and nonexpert users. Both pipelines process raw reads to produce capable files for downstream analyses. MethylStar is the more favourable preprocessing pipeline compared to SINBAD in terms of ease, as it contains a “Quick Run” option to allow all the steps to be run in one go, followed by an e-mail update when this step is finished. However, to impose more specific parameters (e.g., specific Phred scores), SINBAD may be the better option. Another pipeline, MethylPy [109], is similar to MethylStar in its contents; however, MethylStar was able to perform the preprocessing step for 100 single cells in 2225 min, compared to MethylPy, which took 5518 min [112].

### 4.2. Normalisation 

Before biological variation can be inferred, any source of technical variation needs to be removed. Normalisation aims to remove any sources of systemic variation. The normalisation step is essential for downstream analyses, such as principal component analysis and identification of differentially methylated regions [107]. Single-cell transcriptomics have spike-in standards to control for technical noise, yet strong normalisation strategies have not yet been established for single-cell epigenome sequencing. A group attempted to alleviate this by combining approximately 100 single cells to identify peaks via algorithms already in place for bulk sequencing, then looked at each cell to see if these peaks were present [129]. This aggregation method, however, cannot account for missing peaks in single cells with areas of low DNAme levels. Several methods have also been formed by comparing regions that have similar methylation levels in cis-regulatory elements through ENCODE [130].

### 4.3. Data Sparsity

Imputation is used to make statistical inferences to predict the methylation status of CpGs which are not present as a result of dropout events [131]. A dropout event describes the fact that there will be excessive zeroes (meaning no methylation) due to the low sequencing depth. Dropouts also occur when an allele is lost during PCR amplification, meaning we cannot assume biallelic amplification has occurred [107]. This is an important consideration, as it has been shown that many imprinted genes contain methylated cytosines adjacent to G-quadruplexes, which is thought to sterically hinder Taq polymerase, resulting in a dropout event [132]. Such an allelic dropout event can result in one allele being amplified, while the other is lost; in the case of imprinted genes and other mono-allelically methylated genes, false methylation or lack thereof will be observed. 

Imputation allows us to recuperate any lost information, which may lead to the identification of DMRs, subpopulations, and other biologically relevant findings. While it is a necessary step, it should also be approached with caution, as inferring CpG states incorrectly could lead to false biological findings. Traditional bulk methylome sequencing imputation tools would not have to infer as many CpG sites, as there would be far greater genomic coverage. As such, new tools are being developed, which consider the great areas of sparsity in the single-cell methylome data. One group created the MELISA pipeline to address these concerns [110]. The MELISA pipeline largely focuses on two concepts for their imputation strategy: the fact that local methylation profiles influence the epigenetic state of a region more than a single CpG and that many neighbouring cells are also analysed during single-cell analyses. As such, they use a Bayesian hierarchal model which combines the status of neighbouring CpGs in the region and neighbouring cells with similar methylation patterns to infer any missing data. Similarly, DeepCpG uses deep neural networks to predict methylation states of single cells via leveraging of DNA sequence patterns and methylation sites with the state of neighbouring CpGs, both within and between cells [133]. In a comparison performed between MELISA and other imputation tools, it was found that MELISA and DeepCpG had comparable imputation performance to one another, and both outperformed other available methods [133]. The computational overheads of DeepCpG were extensively more demanding than MELISA, however, with a run time of 3–4 days on a GPU cluster system versus 6 h on a small server machine, respectively. However, this reflects that DeepCpG is used to predict genome-wide single CpG methylation states, whereas MELISA focuses on a set of genomic contexts, such as promoters. As such, the choice of which one to use for imputation comes down to whether genome-wide imputation is necessary for the research question or if targeted genomic contexts will be sufficient. Another imputation method similar to MELISA and DeepCpG is Epiclomal [111]. Epiclomal follows the same Bayesian clustering model as MELISA and, while they have very similar imputation methods, MELISA and DeepCpG are able to model spatial variability of neighbouring CpGs, whereas Epiclomal does not. The previously mentioned pipeline SINBAD also has built-in imputation methods; however, they used a simple population methylation mean to replace missing CpG status, which is a far less reliable imputation strategy than that of MELISA and DeepCpG. The genomic context, i.e., whether the CpG is in a CGI, will influence the likelihood of that CpG being methylated; as such, taking the average of a genomic bin, as in SINBAD, is a less reliable method than that of DeepCpG or MELISA. Imputation is a necessary step when preparing data for downstream analyses, especially in a single-cell context where there are many missing values. However, once the values have been filled in, they will be treated as true observations in all downstream analyses but, in reality, they are predicted values. Considering the large amount of missing data, there will be a lot of predicted values; therefore, this should be considered when making biological interpretations from downstream analyses. For the above reasons, using biologically informed algorithms, such as MELISA and Epiclomal, to predict methylation status is most advisable to other methods, such as finding the mean population methylation. 

When considering the implications of imputation as discussed above, combining genomic windows in a sliding window approach could be used which does not require the inference of methylation status. The average methylation of a dictated genomic window can be taken, which can alleviate the data sparsity observed. Of course, a window too large could potentially diminish any signals which may distinguish cells from one another, whereas windows too small will scarcely reduce the data sparsity [107]. 

### 4.4. Downstream Analyses

The commonly emphasised advantages of SCS throughout all the literature are the ability to deconvolute populations of cells for the likes of cell tracing and grouping cells into subpopulations. As such, clustering techniques are one of the most widely used tools in the downstream analyses of all single-cell data. Such clustering requires dimensionality reduction techniques to be employed to visualise the similarities and differences between the single cells [107]. On account of the 28 million CpGs found in the genome, DNAme data has extremely high dimensional data compared to gene expression data, with approximately 20,000 genes in the human genome [107]. Popular dimensionality reduction techniques include the principal component analysis (PCA), t-distributed stochastic neighbour embedding (t-SNE) and uniform manifold approximation and projection (UMAP) graphs. In order to reduce the data sparsity and high dimensionality, one can use the sliding window approach and cluster cells at the whole genome level, as seen in Farlik et al. [51]. Conversely, one could focus on specific genomic regions, such as cis-regulatory elements or repetitive elements of interest, to cluster cells based on their similarities/differences in these regards, another technique shown by Farlik et al. [51]. 

Another great visualisation method for clustering single cells together is a heatmap and dendrogram. This will compare differentially methylated regions between cells across the genome/specific genomic regions. These differences and similarities can be used to trace the clonal evolution of the tumour and identify differentially methylated regions between different subpopulations [107]. The t-SNE and heatmap analyses, although often used in bulk DNAme sequencing, show far higher resolution and insight into the inter-relation between cells in a tumour. The above analyses also could be used to visualise differences between CTCs from the same patient’s tumour, as well as between patients [54]. There are many applications for this, from clustering CTCs from the same patient over time to see the epigenetic developments over treatment, to clustering CTCs from different patients to see if there are correlations between epigenetic signatures in CTCs and stage of disease. Furthermore, pairwise dissimilarity analyses can be performed, which highlight the heterogeneity of the sequenced tumour. Pairwise dissimilarity matrices differ from other clustering methods, as it appreciates the influence of dissimilarities between cells, as well as the similarities [134]. Appreciating both positive and negative correlations between cells helps infer the epigenetic entropy of different cells within the tumour, showing how individual cells’ epigenetic states differ. 

Finally, once subpopulations have been identified with differentially methylated regions, gene ontology (GO) enrichment analysis can then be performed to identify whether differential methylation is potentially associated with functional characteristics [107]. For example, subpopulations may show differentially methylated regions in EMT genes, suggesting they have greater metastatic capabilities, or differential methylation in stem-cell-like genes indicating a possible cancer stem cell population. Gene set enrichment analysis methods specifically for DNA methylation data have been developed, including GOmeth and Goregion, the latter of which is the only method to specifically test enrichment of gene sets for differentially methylated regions [135]. These approaches are advantageous, since gene expression and methylation status at a gene are not necessarily directly correlated.

The downstream analysis is similar between single-cell and bulk, in that similar plots are still produced; however, bulk is assessed via regions, while single-cell is assessed per cell. 

scMethBank offers a simple online tool for researchers to upload their data and obtain visualisations, such as violin and lollipop plots [117]. However, these are for very basic analyses. Predominantly, this website acts as a database for whole-genome methylation profiles in human and mouse single cells, rather than a website for data visualisation. It may be useful to obtain a quick insight into the data before performing further analyses, rather than to produce academic analyses. SINBAD provides a more comprehensive analysis module, which includes dimensionality reduction methods, such as PCA and UMAP, to identify cell clusters. Additionally, it offers a module to identify differential methylation at genes and functional DNA elements amongst the identified cell clusters [113]. Therefore, while scMethBank may be beneficial for a quick preview of what to expect from the data, SINBAD is of far greater scope, providing cell-type classification and identification of differentially methylated regions which may be novel.

## 5. Current Challenges in the Field

The ability to sequence individual cells to uncover cellular heterogeneity at a mono- or multi-modal level is a big jump in the field, yet not without its challenges. There tends to be coverage nonuniformity, sparse data, false positives, amplification biases, and allelic dropout events [136].

### 5.1. Bioinformatic Challenges 

As aforementioned, high-throughput single-cell DNAme studies have CpG coverage of around 5% [77], while low-throughput studies have around 20% genome-wide CpG coverage [82]. This makes it relatively difficult to distinguish cells from one another with large gaps or to infer the epigenetic control mechanisms of that cell with very sparse coverage. A few analysis tools, such as the previously described methylation inference for single-cell analysis (MELISA), have been created to alleviate these issues [110]. 

As mentioned earlier, the popularity of scRNA-seq has resulted in numerous bioinformatic tools to aid in making inferences from single-cell data. Clustering methods for cell population characterisation [137,138] as well as network inference tools [139] have been developed for scRNA-seq techniques. Comparatively, the lack of studies conducted thus far in scDNAme analyses is reflected in the lack of bioinformatic tools.

### 5.2. Experimental Challenges 

On account of the minute amounts of DNA as starting material, the technical noise is substantial. The scarce coverage of processed single-cell epigenomic data requires appropriate normalisation of data, and these high levels of noise need to be accounted for [107]. Another challenge with the minute DNA is the bisulphite conversion process, as previously discussed. It is a harsh process on a small amount of DNA, leading to degradation and, consequently, information loss [50]. Although single-cell methods have been adapted to minimise this (PBAT previously explored), loss of DNA still occurs in the subsequent library preparation. While there are enzymatic alternatives to bisulphite conversion, such as MSRE, such techniques are only capable of cleaving at specific sites—whereas bisulphite conversion allows for the recognition of the single CpG methylation status. Ideally, as long-read sequencing technologies, such as PacBio and ONT, lower their required input and develop improved methylation calling, the above experimental challenges would become null. 

The potential for contamination throughout the process of single-cell DNAme is also very high and may skew results. Any DNA which contaminates samples early in the experiments will be amplified with the cellular DNA and may provide false results. Moreover, the multiple rounds of amplification required means the addition of reagents may lead to further contamination. Negative controls at multiple times point, such as before bisulphite treatment, after and throughout amplification, may help to alleviate these issues [57].

Additionally, occasionally during the cell sorting process, more than one cell can be isolated in a tube, termed a “doublet”. In single-cell research, this is evidently undesirable because it could result in cells that appear to be in transitory or intermediate states [107]. In scRNA-seq, there have been computational tools created to address this issue, such as DoubletFinder [140]. DoubletFinder predicts whether a doublet is present via averaging the transcriptional profile of randomly chosen cell pairs; however, such methods have not been developed for scDNAme contexts. For single-cell DNA methylation studies, sequencing read depth and amount of retrieved CpG sites were analysed as an indicative feature to assess potential amount of doublet cells [73,107]. However, more robust methods that enable accurate identification of doublet cells in single-cell methylome need to be developed for better quantification of single-cell methylomes. 

### 5.3. Challenges in Clinical Implementation 

Another context which provides new challenges is clinical implementation. Theoretically, the use of SCS in a clinical context, particularly in targeted therapy for cancer patients, seems promising. To be able to identify subpopulations that may be treatment-resistant, track tumour progression, and predict prognoses would transform our ability to perform targeted therapy. However, more cost- and time-efficient methods need to be developed that can be easily utilised in clinics. This includes cell preparation and streamlined data analysis pipelines, where only DNAme levels at the loci relevant to the clinical phenotype are shown. Additionally, considering the low genomic coverage and, therein, a high number of predicted methylation states, using this technology to make biological inferences at its current state is not reliable enough to be making clinical decisions for patients. However, all emerging technologies initially have a high cost but, eventually, become very affordable. The cost of single-cell transcriptomic assays has already fallen considerably [141], and single-cell epigenomics will likely follow. Furthermore, SCS can be used to identify biomarkers of previously unattainable populations (therapy-resistant cells/cancer stem cells), which can then be utilised in present clinical diagnostic molecular tests. Therefore, despite the current high cost and a lack of standardised procedures for direct use in the clinic, SCS can still indirectly contribute to improved patient outcomes. 

## 6. Conclusions

The role of epigenetics in cancer is well explored, but the emergence of single-cell technology allows us to review and potentially correct current biological models. This review highlighted the ways in which SCS is an advantageous technique to use to understand such a heterogeneous disease as cancer. The inherent heterogeneity harboured within and between tumours often leads to therapy resistance or disease relapse. Not only this, but it makes cancer a distinctly difficult disease to research. The divergence of cells needs to be appreciated to further our understanding of the disease, making single-cell sequencing a favourable technique. This appreciation has been shown in the recent surge of single-cell RNA analyses; however, there are still few single-cell methylome analyses in cancer due to the difficulties of the current protocols. However, as these protocols improve, we expect to also see a surge, given the epigenomes’ great influence on the cancer genome. Single-cell technologies offer the detection of heterogeneous populations, the ability to analyse small subpopulations of cells, and the ability to delineate cell maps. Although there is much room for improvement, single-cell DNAme sequencing has exciting prospects and is sure to become common practice in coming years. 

## Figures and Tables

**Figure 1 cancers-14-06171-f001:**
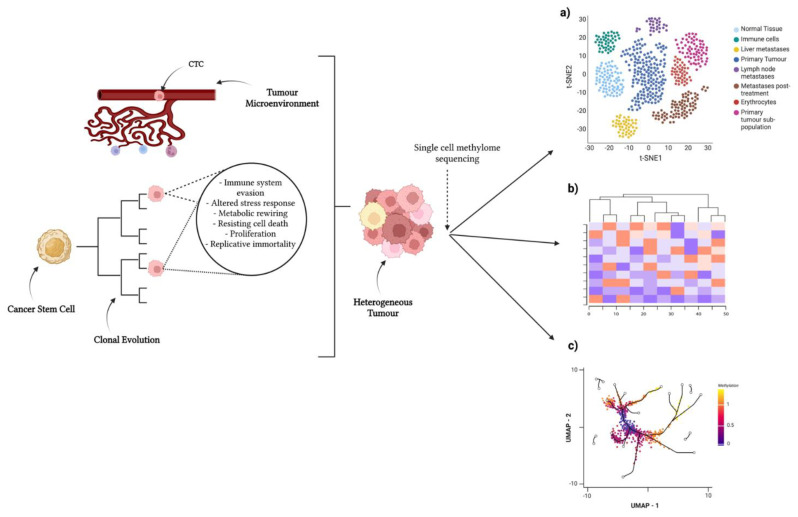
Single-cell approaches to identify heterogeneity in tumour population. Tumour-initiating stem cells develop in tissue of origin. These undergo branched tumour evolution, acquiring random mutations and epigenomic alterations. The tumour microenvironment influences heterogeneity via physical and chemical signals. A combinatorial effect of these concepts induces a highly heterogeneous tumour, as well as circulating tumour cells (CTCs) disseminating from said tumour. Single-cell methylome sequencing of tumours can provide insight into subpopulations/differing cell states by clustering cells: (**a**) clustering via t-SNE, (**b**) clustering via UMAP, and (**c**) cell lineages/differentially methylated regions via heatmaps. Created with https://biorender.com/ (accessed on 25 July 2022).

**Figure 2 cancers-14-06171-f002:**
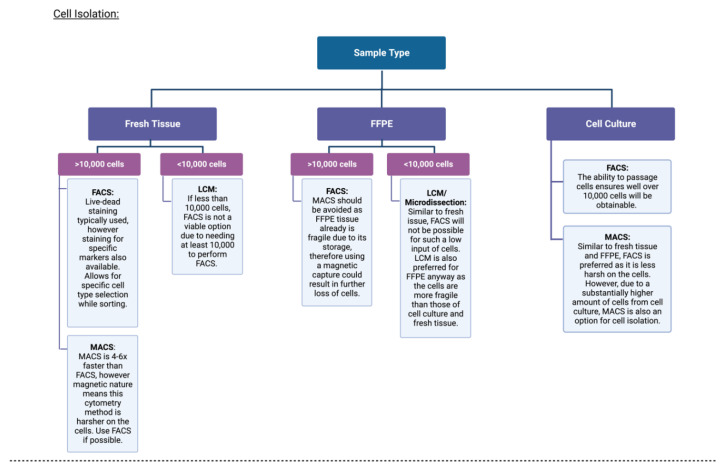
Single-cell isolation and experimental approaches for single-cell DNA methylation analyses. Created using https://biorender.com/ (accessed on 25 July 2022).

**Table 1 cancers-14-06171-t001:** Examples of single-cell methylome sequencing experiments in cancer.

Cancer Type	Genome-Wide or Gene-Specific	Findings	Year Published	References
HL60 (acute promyelocytic leukemia cell line) and K562 (erythroleukemia-derived cell line)	Genome-wide	First implementation of single-cell WGBS	2015	[51]
Hepatocellular carcinoma	Genome-wide	Identification of subpopulations within tumour	2016	[52]
**Metastatic breast** **cancer (mBC) and metastatic castration-resistant prostate cancer (mCRPC)**	CDH1 and miR200 promoters.	CTCs from same patient displayed heterogeneous methylation patterns. Different methylation patterns at these promoters in mCRPC vs. mBC CTCs suggesting differentially regulated miR200 loops in these two tumour entities.	2017	[37]
**Colorectal cancer**	Genome-wide	Sub-lineages identified in patients found metastases at multiple sites had a common origin	2018	[38]
**Chronic Lymphocytic Leukaemia**	Genome-wide	Subpopulations preferentially expelled from lymph nodes after treatment	2019	[33]
Lung Adenocarcinoma	Genome-wide	Global methylation heterogeneity amongst tumours associated with the progression of LAC	2021	[53]
Lung Cancer	Genome-wide	Unique CTC DNA methylation signature distinguished it from the primary tumour	2021	[54]
6 Cancer Types: Prostate, Colon, Small cell lung, Lung Adenocarcinoma, Breast, and Gastric	Genome-wide	Potential to identify tumours of origin for CTC based on methylome profiles. Report diverse evolutionary histories of CTCs	2021	[47]
KG1a Acute Myeloid Leukaemia	Transposable elements: SINE Alu	TEs as a surrogate for predicting single-cell global DNA methylation. Method has greater alignment and costs 3-fold less than scBS-seq	2022	[55]

**Table 2 cancers-14-06171-t002:** Single-cell methylome sequencing techniques. Adapted from Kashima et al. (2020) [72].

Method	Key Features	Year of First Study	References
Single-cell reduced representation bisulphite sequencing (scRRBS)	Bias in regions with high CpG density, limited coverage in regions with low CpG density	2013	[69]
	Cost-effective		
Single-cell bisulphite sequencing (scBS-seq)	Single base resolution High cost	2014	[73]
	DNA degradation.		
Quantitative RRBS (Q-RRBS)	Incorporated UMIs for PCR-duplicate removal	2015	[71]
Single-cell AbaSI sequencing (scAba-seq)	Low false-positive rate	2016	[74]
	Distinguishes between 5hmC and 5mC		
	No chemical degradation		
Single nucleus methylcytosine sequencing (snmC-seq2)	Reaction occurs within the nucleus Single-strand library preparation	2018	[75]
Single-cell combinatorial indexing for methylation analysis (sci-MET)	Lower coverage but higher throughput relative to other methods	2018	[59]
Microfluidic diffusion based RRBS (MID-RRBS)	Diffusion-based reagent exchange allows for minimal loss of DNA. Microfluidic device allows for multiple cells to be done in parallel.	2018	[62]
Single-cell methylation-sensitive restriction enzyme sequencing (scMSRE)	Analysis limited to methylation at restriction sites No chemical degradation scCGI	2021	[76]
Single-cell transposable element sequencing (scTEM-seq)	Uses transposable elements as surrogates to predict single-cell global methylation	2022	[55]

**Table 3 cancers-14-06171-t003:** Currently available bioinformatic analysis pipelines for single-cell DNA methylation sequencing.

Analysis Category	Name of Pipeline and Year Published	Environment	Features
Preprocessing	MethylPy (2015) [109]	Python	Processes raw reads through to methylation state. Combines data from adjacent cytosine for dealing with low coverage data
Imputation	MELISA (2019) [110]	R	Uses information from neighbouring CpGs and from neighbouring cells with similar CpG patterns to predict missing CpG methylation states. Also uses Bayesian clustering to identify subsets of cells based on epigenetic state
Imputation	Epiclomal (2020) [111]	R and Python	Simultaneously clusters sparse single-cell DNAme data and imputes missing values
Preprocessing	MethylStar (2020) [112]	Python	Contains a “quick run” option that streamlines all preprocessing steps, including trimming, alignment, removal of duplicates, and methylation calling
Overall Analysis	SINBAD (2021) [113]	R	Contains 5 modules consisting of pre-processing, mapping, methylation, dimensionality, and gene signature profiling
Downstream Analyses	EpiScanpy (2021) [114]	R	A scRNA-seq workflow adapted for sc-ATAC and sc-DNAme analyses
Preprocessing	scMET (2021) [115]	R	Hierarchical Bayesian model designed to overcome data sparsity. Also performs differential methylation and variability analyses
Downstream Analyses	scMelody (2022) [116]	R and Python	Consensus-based clustering model that takes into account distance relationships between cells to improve the identification of subpopulations
Downstream Analyses	scMethBank (2022) [117]	Online	Provides curated metadata of 8000+ samples of different cell types and states. Provides online tools for simple and practical downstream analyses such as lollipop plots, DMR annotation and enrichment analysis

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
