# Peer review of "Single-Cell DNA Methylation Analysis in Cancer"

_cancers, 2022, doi:10.3390/cancers14246171_

Round 1
Reviewer 1 Report
Thank you for the opportunity to read the interesting manuscript called Single-Cell DNA Methylation Analysis in Cancer. The review is interesting and covers a broad spectre of subjects within single-cell sequencing and the methods related to this. The manuscript is well-written and the structure is logical.
Reviewer 2 Report
O’Neill et al have written a review on the single-cell DNA methylation analysis in cancer. This is particularly important as single-cell analysis of transcriptomic and particularly epigenomic information has made possible only recently due to technical advances. Given the dynamic nature of a cancer’s epigenome and genome, specific cell populations may have novel transformations, advancing the progression or aggression of the cancer/tumor. If these cells are the minority of the population and sequenced under “bulk sequencing,” they may be unfortunately considered non-significantly altered genes therefore missed during bioinformatic analysis, yet they very well may be clinically relevant. Single-cell sequencing allows for more thorough analysis of sub-populations of cancer cells, likely moving towards the gold standard for clinical analysis in the near future, particularly if a more personalized medicine approach is adapted. Therefore O’Neill et al have been very timely in the writing of this review as new technical advances in this field are worthy of discussion.
However, there are several points in the paper where the language shifts to a more informal style of writing, several important abbreviations are not defined, and lack of citation. Unfortunately these points are numerous enough to become a major issue. The following are my comments to the authors addressing these specific points:
Major comments:
I would recommend an abbreviation key or legend. Some of the abbreviations may become confusing to the reader, particularly when similar abbreviations like SCS and CSC are discussed at the same time in a paragraph. Additionally, several of these abbreviations have not been defined – see minor comments. Given the technical quality of this paper, I think it may be helpful, especially to readers who are not experts in the field, but are instead attempting to increase their knowledge base (graduate students, clinicians, etc.).
In 3.2, the experimental approaches section, there is a great deal of discussion about various forms of bisulfite conversion, however it may be helpful for readers to understand what exactly bisulfite conversion does – for example, converting a methyl-CpG to a uracil. I would also suggest briefly discussing why methylated DNA immunoprecipitation (MeDIP) is not a viable technique in single-cell sequencing, despite regularly being used in DNAme analysis. This would probably fit in well in paragraph 2 where you discuss non-conversion techniques.
There are several areas that lack appropriate citation of information. I would highly recommend that all authors do a much more thorough internal review of the manuscript prior to approving it to be submitted. For example, there is not a single citation between lines 401-416. This is absolutely needed. There are several areas sporadically through the paper where there is a lack of citation, not just in a summarization of discussion (ex: 426-445; 459-464; 467-49; 729-747 and several others areas). This is a serious issue of academic integrity which has unfortunately lead to my decision to recommend to reject the manuscript.
Minor comments:
First sentence of the introduction (line 40) starts off very confusing as many, if not the majority of readers may not be familiar with this specific publication and its descriptors. It may instead be better to describe how epigenetics used to be a largely mysterious field or largely misunderstood in regards to how it can contribute to cancer.
In line 64, it may be helpful to define CpG islands. You do state that CpG sites are “usually concentrated in CpG islands” but don’t define these islands specifically as areas with higher frequencies of CpG sites.
Several abbreviations have not been defined:
In line 175, CRC (presumably colorectal cancer), has not been defined as what it is an abbreviation of.
Similarly, in line 172, EMT has not been defined either. Not all readers will be aware that this is epithelial-to-mesenchymal transition.
In figure 2, FFPE is not defined (should probably discuss in the paragraph about LCM starting at line 265).
Line 330, 5hmC is not defined nor is 5mC – you do discuss in that sentence that hydroxymethylation is implicated in several cancers. It may be beneficial to actually define and discuss the relevance in the beginning of the review.
Line 366 - should probably define gDNA as presumably genomic DNA
In line 183, the gene name MGMT should be italicized to ensure that it is clear that the authors are referring to a specific gene
In line 222, what is meant by harsh specifically? Might be helpful to be more specific. I assume this is referring to DNA degradation as a result of the process. In the context of SCS, this degradation may actually make it an unviable technique. “Harsh” is also used in line 255 referring to the MACS process. I assume this means that a higher number of cells lyse, but again, it may be helpful to the reader to be more specific.
Line 369-380 converts to second person “you.” This is informal and not commonly considered appropriate for scientific writing. This also occurs in line 488, 551, 699, 703, and 708. The reason is that “you” is highly ambiguous to a general reader where the audience of this paper can include anyone from a primary investigator at a research institute to a clinician to an undergraduate molecular biosciences student. Not all will be “aiming to infer a correlation between gene promoters and expression.”
The text is very small in figure 2 if printed out, bordering on illegible. Please request that the editors make this figure larger when resubmitting.
The “experimental approach” figure (line 390) does not have a figure label or figure description. The same text illegibility issues discussed in figure 2 also apply here.
Lines 594-603 – why is this all in italics? Was this an issue with the editor formatting?
Reviewer 3 Report
In this review manuscript the authors discuss the use of different technologies available for assessing DNA methylation from single cells. The authors have strongly focused on applications in cancer research and diagnosis, and provide a thorough review of the literature and a detailed discussion of the bioinformatic processes. The strengths and limitations of the different approaches and how these are best utilized in biological contexts has also been well discussed. The manuscript is of a high grammatical standard, contains few errors, and is of an acceptable standard for publication.
I have listed my minor comments here:
1. Figure 1A) Figure legend on the image (labels) requires modification and is not really appropriate for the image
2. Abbreviations are some what liberally used, generally if unless used more than five times, they should generally be spelled in full. For examples, CLL and TME abbreviations are probably not necessary
3. EMT abbreviation, does not appear to have been spelled in full. Epithelial-mesenchymal transition?
4. Line 183, section 2.3. MGMT gene should be italicized
5. Lines 216 -220 could be revised for clarity
Round 2
Reviewer 2 Report
I thank the authors for their extensive revisions. It is my hope that with these changes, this review will be a much stronger contribution with the field. I still do see some minor revisions needed (which would likely be caught by journal staff when processing the manuscript for final publication), but with those edits, I will recommend to accept the manuscript.
line 69 – this was changed from 1.1 to 1.2 however line 93 is also 1.2. Should DNA methylation stay 1.1?
Line 104-108; 898-906 are not in the same paragraph format.
Line 243 – Sentence has two instances of “in a clinical context” – should probably read instead “In a clinical context, it has also been proposed that SCS could provide…”
